# Ribosome Pausing at Inefficient Codons at the End of the Replicase Coding Region Is Important for Hepatitis C Virus Genome Replication

**DOI:** 10.3390/ijms21186955

**Published:** 2020-09-22

**Authors:** Gesche K. Gerresheim, Carolin S. Hess, Lyudmila A. Shalamova, Markus Fricke, Manja Marz, Dmitri E. Andreev, Ivan N. Shatsky, Michael Niepmann

**Affiliations:** 1Inst. of Biochemistry, Medical Faculty, Justus-Liebig-University, 35392 Giessen, Germany; Gesche.Gerresheim@gmx.de (G.K.G.); Carolin.S.Hess@med.uni-giessen.de (C.S.H.); 2Inst. of Virology, Faculty of Veterinary Medicine, Justus-Liebig-University, 35392 Giessen, Germany; ludmilashalamova@gmail.com; 3Genevention GmbH, 37079 Göttingen, Germany; markus.fricke2@googlemail.com; 4RNA Bioinformatics and High Throughput Analysis, Faculty of Mathematics and Computer Science, Friedrich Schiller University Jena, 07743 Jena, Germany; manja@uni-jena.de; 5Lomonosov Moscow State University, Belozersky Inst. of Physico-Chemical Biology, Moscow 119234, Russia; cycloheximide@yandex.ru (D.E.A.); ivanshatskyster@gmail.com (I.N.S.); 6Shemyakin-Ovchinnikov Institute of Bioorganic Chemistry, Moscow 117997, Russia

**Keywords:** HCV, translation, ribosome pausing, rare codon, Wobble codon, replication, minus strand, negative-strand, antigenome

## Abstract

Hepatitis C virus (HCV) infects liver cells and often causes chronic infection, also leading to liver cirrhosis and cancer. In the cytoplasm, the viral structural and non-structural (NS) proteins are directly translated from the plus strand HCV RNA genome. The viral proteins NS3 to NS5B proteins constitute the replication complex that is required for RNA genome replication via a minus strand antigenome. The most C-terminal protein in the genome is the NS5B replicase, which needs to initiate antigenome RNA synthesis at the very 3′-end of the plus strand. Using ribosome profiling of cells replicating full-length infectious HCV genomes, we uncovered that ribosomes accumulate at the HCV stop codon and about 30 nucleotides upstream of it. This pausing is due to the presence of conserved rare, inefficient Wobble codons upstream of the termination site. Synonymous substitution of these inefficient codons to efficient codons has negative consequences for viral RNA replication but not for viral protein synthesis. This pausing may allow the enzymatically active replicase core to find its genuine RNA template in *cis*, while the protein is still held in place by being stuck with its C-terminus in the exit tunnel of the paused ribosome.

## 1. Introduction

Hepatitis C virus (HCV) is an enveloped RNA virus classified in the family *Flaviviridae*, genus *Hepacivirus*. The virus comes as a mixed lipo-viro particle and mostly infects human liver hepatocytes by binding to a variety of receptors [1,2]. In many cases, HCV infection is not eliminated by the immune system but becomes chronic, often also leading to liver cirrhosis and liver cancer (hepatocellular carcinoma; HCC) [3,4,5,6]. The virus usually replicates slowly and does not lyse the host cells, but it sneaks out rather discreetly by using the cellular export pathway for lipoprotein particles [7,8,9,10,11,12,13]. The resulting liver-derived mixed lipo-viro particles serve to hide the virus from the immune system and to enhance uptake of the viral particles by the next hepatocyte host cell by the use of apolipoproteins [6,10,13]. Moreover, HCV has developed several strategies to counteract the immune response [14,15,16], including the use of cellular long non-coding RNAs [17,18]. Chronic HCV infection is often recognized only accidentally, e.g., during hospitalization, a reason why virus spread in the population is often unnoticed [14,15]. Treatment of HCV has been largely improved in the last years by the development of highly efficient direct-acting antivirals (DAAs), but such treatment is still extremely expensive and, therefore, used only after trying other treatment options first [19,20,21]. Although a lot has been learned about the replication of the virus, many details of HCV replication remain to be elucidated [20]. Therefore, there is still an urgent need to better understand the molecular biology of HCV replication.

The HCV genome is a positive-sense RNA (i.e., of mRNA polarity) of about 9600 nucleotides (nts) in length [22]. After entry into the cytoplasm of the cell, the plus-strand genome can be directly translated by virtue of the internal ribosome entry site (IRES) (Figure 1) [23,24]. The product of the viral polyprotein open reading frame (ORF) is a precursor protein that is co- and posttranslationally processed into the mature gene products [8,25,26,27], including the structural proteins with the core protein acting as a nucleocapsid protein, and the envelope proteins that are incorporated in the viral lipid envelope. The non-structural (NS) proteins include all other activities that are required for virus replication but are usually not included in the viral particles. These proteins include the p7 ion channel; NS2, which is involved in protease assembly; the NS3/NS4A protease/helicase; NS4B, which is involved in membrane organization; NS5A that is involved in RNA replication and assembly, and finally—at the very end of the protein-coding sequence—the NS5B protein, the RNA dependent RNA polymerase (RdRP) or replicase [27]. Viral RNA genome replication takes place in the cytoplasm in the so-called membranous web, membrane compartments that are recruited from the endoplasmic reticulum and provide a protected and structurally organized environment for the viral replication and assembly machinery [13,27,28].

The viral translation is regulated cap-independently by the IRES [23,24], and the RNA *cis* elements involved in the regulation of virus genome replication largely reside in the untranslated regions (UTRs) at the genome ends. In the 5′UTR, these *cis* elements include the 5′-terminal stem-loops (SL) I and II and the region in between that is used for binding of the liver-specific microRNA (miR) 122. In contrast to the usually negative action of microRNAs on normal cellular mRNAs [32,33], miR-122 positively regulates HCV genome translation, replication, and stability [23,30,34,35,36]. In the 3′UTR, a variable region of yet unknown function is followed by a single-stranded poly(U/C) tract and the highly conserved so-called 3′X region. The poly(U/C) tract is required to have a certain minimum length for allowing virus replication [37,38,39]. The terminal 3′X region can dynamically fold either into three or into two stem-loops [30,31]. 

In addition, a variety of regulatory elements is found, in particular, in the NS5B coding region at the end of the coding region [23,30,31,40,41,42,43,44]. The most prominent of these *cis*-regulatory RNA elements in the NS5B coding region is the *cis* replication element (CRE) (also called 5BSL3.2) that is required for replication [45]. Together with its upstream and downstream neighbor structures—5BSL3.1 and 3.3—the 5BSL3.2 can bind the NS5B replicase [46,47,48,49]. However, this RNA region can also bind the small ribosomal 40S subunit [50], and it has been reported to be involved in translation regulation [51]. The 3′UTR is—in addition to its role in replication—involved in translation stimulation [23,52,53], and it can also bind the ribosomal 40S subunit [54]. Moreover, the CRE/5BSL3.2 and the 3′X interact with each other by virtue of a so-called “kissing loop” interaction that involves the apical loop of the CRE and the SL 2 of the 3′X region (when present in its three-stem-loop form) [31,39,55]. Thus—besides many other possible long-range RNA-RNA interactions [31,43]—this network of interactions involving the CRE region, the 3′X region, the NS5B replicase protein, and the ribosomal 40S subunit could be considered to be involved in the regulation of the transition from genome translation (that is required to produce the viral replication proteins) to the initiation of viral RNA replication. 

In light of these facts, we aimed at analyzing the possible role of translation events at the end of the NS5B protein-coding region and their possible role in regulating the initiation of RNA genome replication. We found that two ribosomes are pausing at the translation termination site and at the next ribosome occupied position directly upstream. This pausing is caused by the use of rare, inefficient codons, and it serves to enhance replication. We hypothesized that this ribosome pausing provides enough time for the NS5B replicase emerging from the ribosome to find its genuine template, the 3′-end of the viral genome, in *cis*. 

## 2. Results

### 2.1. Analysis of Ribosome Occupancy of the HCV Genome 

In order to investigate possible ribosome pausing in the NS5B region, we analyzed the ribosome occupancy profile of the HCV plus-strand genome (Figure 2A). Full-length infectious HCV RNAs (Jc1 clone) [56] were transfected into Huh-7.5 hepatoma cells and allowed to replicate for 6 days to achieve full infection of the cells. At day 6, cells were harvested, and ribosome profiling and transcriptome analyses were performed, as described previously [57]. Ribosome occupancy reads on the HCV plus-strand genome are shown in Figure 2B (upper panel). As a control, in the ribosome profiling of the minus strand, we see virtually no reads (Figure 2B, lower panel), consistent with the notion that the plus but not the minus strand is occupied with ribosomes for translation [23,29,30]. As a further control, transcriptome reads were mapped to the HCV plus and minus strands (Figure 2C). In the transcriptome reads, low level of minus-strand reads compared to plus-strand reads are detected (Figure 2C), which is consistent with the ratio of plus to minus strand abundancies of about 5:1 to 15:1 in HCV replicating cells [22,58]. 

Interestingly, we do not see a prominent 80S ribosome peak at the HCV start codon (position 341). This is consistent with our previous finding that the attachment of the small ribosomal 40S subunit to the HCV IRES is slow, while the complete 80S ribosome formed after 60S subunit joining leaves the initiation site without substantial delay [35]. In addition, we see a slight increase in ribosome occupancy in the first about 200 nucleotides of the coding sequence (positions 341 to 550). This effect is reminiscent of the so-called “ramp” phenomenon, which means that ribosomes accumulate in a queue at the beginning of the coding sequences of cellular mRNAs [57,62]. 

Over the entire genome, we see many peaks, with three prominent peaks (by peak height) near HCV positions 1980 and 3100 and at the polyprotein stop codon. The strongest peak located at the HCV coding sequence stop codon around position 9440 (Figure 2B) particularly caught our attention. This peak has no prominent counterpart in the transcriptome reads (Figure 2C). Moreover, metagene profiles of cellular mRNAs do not show such significant accumulation of ribosomes at the stop codon, which rules out the possibility of experimental artifacts of genome-wide stop codon reads accumulation [57]. Thus, we suspected that this peak might be actually caused by higher ribosome occupancy on the HCV RNA at this position, and we wondered if this ribosome peak at the stop codon might have something to do with a possible regulation of the switch from translation to replication. Therefore, we further focused on this peak. 

### 2.2. RNA Structure Analyses

Considering possible mechanistic reasons for this ribosome pausing event at the end of the NS5B coding region, we first checked for RNA secondary structures, which could be considered to slow down ribosome movement by acting as a “roadblock”, like in ribosomal frameshifting [63,64,65,66]. In such a case, a pausing ribosome would be found directly upstream of a strong RNA secondary structure. Therefore, we analyzed the predicted double-strand (ds) probability of the HCV RNA genome (Figure 2D) in order to identify very strong RNA secondary structures. However, at first sight, ds probability largely fluctuates around an average value, with only a few remarkable characteristics. Firstly, the region just downstream of the HCV start codon at position 341 appears to be of low double-strand probability, consistent with the need for ribosome assembly at the start site. For comparison, the miR-122 target sites in the HCV NS5B coding region (indicated by small vertical boxes in Figure 2A,D) appear to be located at only moderate local minima of ds probability (Figure 2D). 

Secondly, the ds probability profile does not show a peak at the end of the coding region, which is remarkably higher than other regions, and that could be suitable for explaining the strong ribosome occupancy peak at the end of the coding region. Nevertheless, at the position of the CRE element (stem-loop 5BSL3.2, see also Figure 1), which has only short ds RNA stretches but also a bulge and a long loop [30,31,43,45,48,55], the ds probability is average (“3.2” and white arrow in Figure 2D). In contrast, at the position of the stem-loop 5BSL3.3, which has a long continuous stem (see Figure 1) [30,31,43], we see a sharp local peak in ds probability (“3.3” and grey arrow in Figure 2D). However, the position of this 5BSL3.3 is not located downstream of the ribosome peak seen at the stop codon but overlaps with this ribosome peak. Thus, the stem-loop 5BSL3.3 cannot be argued to cause ribosome pausing by acting as a “roadblock” immediately downstream of a pausing ribosome. 

Another type of RNA secondary structure analysis can be used to measure the presence of so-called RNA secondary structure classes in protein-coding RNA regions [60]. For this kind of analysis, RNA sequences are folded in silico, and the RNA secondary structures are inspected for the overlapping open reading frame and classified according to how opposing codons in the RNA secondary structure are predicted to pair with each other. For example, opposing codons can occur in the RNA structure in a way that codon position 1 of one codon is pairing with the codon position 1 of an opposing codon (see Figure 2E, small insert at the left, and [60]). This pairing arrangement is then classified as RNA secondary structure class c1 (Figure 2E). This class c1 with the pairing of codon position 1 with codon position 1 of an opposing codon usually is slightly counter selected in natural mRNAs since the nature of encoded amino acids is largely defined in the first position of a codon [60], and such an arrangement would require the unlikely co-variation of two distant amino acid positions just in order to allow for a certain RNA secondary structure. However, as primarily the amino acid sequence is driving the evolution of the coding sequences, the RNA secondary structure class c1 is, on average, slightly under-represented in coding sequences [60]. Accordingly, in RNA secondary structure class c2, position 2 of one codon is required to pair with position 2 of an opposing codon, and in the class c3, opposing codon positions 3 pair with each other (see Figure 2F,G, small inserts at the left, and [60]). Codon position 2 is largely determining if the encoded amino acid is hydrophobic or not. Therefore, RNA secondary structure class c2 is under-represented in RNA secondary structures that overlap with conserved hydrophobic protein regions, in order to avoid the need for amino acid co-variation. 

For the above reasons, these constraints lead to a bias in the sequences available for the RNA secondary structure: in protein-coding regions containing functional RNA secondary structures, which—on their own—underlie a selection pressure, RNA structures falling into the RNA secondary structure classes c1 or c2 tend to be under-represented [60]. In contrast, the primary sequence underlying an RNA structure that is present in a coding region and that is under selection pressure (i.e., that may have a function) must meet the requirements of the RNA secondary structure, mainly by changes in the 3rd codon position [60]. 

Inspection of the HCV coding sequence analyzed for these RNA secondary structure classes revealed that there are certain conserved highs and lows in the protein-coding HCV RNA sequence that are conserved in several HCV isolates. Most strikingly, we found a very strong under-representation of class c2 at the position of the stem-loop 5BSL3.3 at the very end of the NS5B coding region (Figure 2F), which is compensated by a very strong over-representation of class c1 at this position (Figure 2E). Inspection of the stem of the 5BSL3.3 structure reveals that this RNA secondary structure indeed is an uninterrupted stem structure of class c1 (see below), and class c2 structures are not present in this stem-loop. This is consistent with the fact that the protein-coding sequence of the membrane anchor helix at the very end of the NS5B coding sequence is highly hydrophobic (Figure 2H). The above results also show that the analysis of these RNA secondary structures can at least provide supportive evidence if a conserved uninterrupted stem-loop in a coding region may be of functional importance on the RNA structure level [60]. 

Taken together, RNA secondary structure class analysis provides evidence for the conclusion that the sequence and secondary structure of the 5BSL3.3 are very conserved, pointing to the possible importance of this RNA region in HCV replication. Even though the strong 5BSL3.3 RNA secondary structure could be considered to slow down ribosome movement, its structure is present in an unfolded state when the sequence is covered by the ribosome. In conclusion, other mechanisms must contribute to the ribosome occupancy peak upstream of the stop codon region. 

### 2.3. Ribosome Pausing at the HCV Stop Codon Region

We then had a closer look at the ribosome peaks around the NS5B stop codon (Figure 3). At higher resolution, the broad and high peak at the HCV stop codon at the end of the NS5B polymerase coding region seen in Figure 2B resolves into two largely separate peaks (Figure 3A,B, Appendix A). Each of these two peaks has higher read counts than any other peak in the profile, while the corresponding low peaks in the transcriptome profile (Appendix A) exclude sequencing bias as a cause. 

Each ribosome peak covers about nine triplets or slightly more (Figure 3A and Appendix A). The downstream peak “1” centers on the stop codon, with many read lengths of 27 nucleotides. The upstream peak “2” centers on a valine codon (HCV position 9407, white arrow in Figure 3A). In addition, the read distribution shows that some reads are located between peak 2 and peak 1, suggesting that the ribosomes first sit on peak 2 and then slide over from peak 2 to peak 1 (Figure 3B and Appendix A). A general change in cellular conditions for translation termination can be excluded since the ribosome occupancy of cellular mRNAs essentially does not change during HCV replication [57]. In conclusion, these two peaks at the HCV stop codon may reflect a special situation at the HCV translation termination site.

The strong stem-loop 5BSL3.3 structure overlaps with ribosome peak 2 (Figure 3A). Thus, this stem-loop could be hypothesized to contribute to slow down approaching ribosomes, but the actual pausing at peak 2 cannot be explained by the 5BSL3.3, which is unfolded in the ribosome peak 2. Therefore, we considered other possible reasons for ribosome pausing. Particular “pausing” or “stalling” peptide sequences in a translated protein can cause ribosome pausing by the interaction of selected amino acids with the ribosome exit tunnel [69,70]. However, we could not identify such “pausing” sets of amino acids in the NS5B protein sequence around position 9400. 

The third option for slowing down the ribosome movement would be the use of “slow” or “rare” codons, which are decoded with low efficiency [71]. Often, such codons are Wobble codons that give rise to slow translation [72,73,74], in particular also including rare leucine codons [75]. Actually, an inspection of the codon usage at positions 9392, 9398, 9407, and 9413 (Figure 3C,D and Appendix A) revealed that for the leucine and valine codons, very often CUA or GUA are used, respectively, indicating that the use of slow Wobble pairing and the possible depletion of the cognate Wobble tRNA could contribute to slow decoding and cause ribosome pausing [71,75]. Analysis of these codons (Figure 3C,D) shows that the valine codon at position 9407, which is in the center of ribosome peak 2, is a rare codon (marked by a diamond). Upstream of peak 2, there are three more codons of very rare usage (marked by triangles and a square). Downstream of the ribosome peak 2 centered valine, two more rare codons are found—one for valine (marked by a circle) and one for leucine. Correspondingly, the ribosome read profile (Figure 3B and Appendix A) shows that ribosomes may slide slowly into the position of peak 2, perhaps slowed down by the three upstream rare leucine codons. Then, the ribosomes stay on the rare valine codon in peak 2. After that, they slowly slide towards the stop codon, again slowed down by the rare valine and leucine codons between the two peaks. 

Analysis of the conservation of these codons among HCV isolates from all genotypes shows that those codons, which are in or upstream of the center of ribosome peak 2 (diamond, position 9407), are strongly conserved to be rare codons among HCV isolates (Figure 3D). The codon in the center of peak 2 is conserved to be rare in 86% out of 102 representative HCV isolates from all HCV genotypes and nearly all subtypes [43]. Even more strikingly, the two rare leucine codons at positions 9392 and 9398—preceding the valine codon in the center of peak 2—are strongly conserved to be rare codons in 101 or 96, respectively, out of 102 HCV isolates. The upstream codon at position 9380 is also conserved to a high extent to be a rare codon. From these findings, we conclude that these rare codons may be conserved for a certain function, perhaps to slow down protein synthesis to give some protein regions enough time for proper folding [76,77]. For example, during the synthesis of cellular membrane proteins, translation can be slowed down by rare codons to give the signal recognition particle (SRP) enough time to recognize the candidate protein´s N-terminal hydrophobic signal sequence (reviewed in [72]). In contrast, the codon between the two ribosome peaks at HCV position 9425 is not conserved to be a rare codon (Figure 3D). 

### 2.4. A Replicon System for the Analysis of Rare Codon Function

In order to analyze the possible function of the above rare codons in HCV translation and replication, we designed a replicon system that contains the HCV genome ends required for replication as well as the NS3-NS5B replication protein cassette (Figure 4A,B). Translation of the NS3-NS5B replication proteins is driven by the IRES of encephalomyocarditis virus (EMCV). At the replicon´s 5′-end, the HCV 5′UTR was inserted. It is important to note that the replicon RNAs do not contain a selectable marker like an antibiotic resistance gene. Thus, these RNAs rather constitute “RNA synthesis template RNAs”, which are able to amplify by minus and plus-strand synthesis but in the absence of a selectable marker. For sensitive detection of HCV protein production, a HiBiT tag (amino acids VSGWRLFLLIS) [78,79] was inserted in the protein-coding sequence. As an insertion point for the HiBiT tag, together with upstream and downstream glycine-serine linkers, we chose the C-terminal sequence of the NS5B replicase between the codons for arginine (R) 543 and leucine (L) 544. This insertion point is located about 20 amino acids (AAs) upstream of the hydrophobic membrane anchor of the NS5B polymerase [80]. Thereby, the non-conserved R–L junction is supposed to be located in an α-helical turn positioned towards the solvent [80], and we hypothesized that a HiBiT insertion in this position could still yield a functional NS5B polymerase. 

After transfection of these replicon RNAs into Huh-7.5 hepatoma cells, we analyzed HCV protein expression by measuring the HiBiT tag after 4, 8, 24, 48, and 72 h. A “GND” NS5B polymerase deficient [81] replicon (Pol −) was used as a control. Since progeny plus strands appear earliest after 12 h [82], early HiBiT expression at 4 and 8 h can be regarded as a primary translation of transfected plus-strand RNAs, whereas HiBiT expression after 24 h and later additionally includes replicon RNA amplification and translation of progeny plus strands, with the NS5B polymerase deficient replicon again showing HiBiT expression only from the transfected (but not replicating) RNA. 

Figure 4C shows that HiBiT expression is maximal at 4 and 8 h after transfection. HiBiT translation decreases at 24 h after transfection and then drops to undetectable levels at 72 h due to degradation of the transfected RNAs in the absence of active replication, with no significant differences between the wild-type (wt) NS5B polymerase (Pol +) and the replication-deficient mutant (Pol−). Thus, the HiBiT tag insertion in the NS5B coding sequence obviously is deleterious for polymerase activity, and the observed HiBiT tag expression is only due to the translation of the transfected RNAs. 

In contrast, when we inserted the HiBiT tag upstream of the NS3 gene segment (Figure 4D), HiBiT tag expression is similar for the wt and polymerase-deficient replicons at 4, 8, and 24 h after transfection, whereas after 48 h, and in particular, after 72 h, the wt polymerase replicon strongly increases HiBiT expression, while HiBiT expression from the polymerase-deficient replicon just declines further (Figure 4E). Thus, the HiBiT insertion is not deleterious for NS5B and replication complex activity. HiBiT expression at early time points (4, 8, and still 24 h) indicates the translation of only the transfected RNA molecules. In contrast, increasing HiBiT expression at later time points (48 and 72 h) indicates additional HiBiT expression from newly made progeny plus strands after HCV replicon amplification by ongoing minus and plus-strand production. Within the first 72 h, the expression of the HiBiT tag from the transfected RNAs does not yet show a strong exponential increase. However, those RNAs that carry an active NS5B polymerase catch up and start to replicate, which could be expected to continue at later time points (please see also below). In contrast, those RNAs that carry an inactivated NS5B polymerase [81] are degraded. This replicon system (Figure 4D) was then further used for the analysis of the effects of the rare codons in the NS5B coding region on translation and replication. 

### 2.5. The NS5B Rare Codons Are Important for HCV Genome Replication but Not Translation 

In order to analyze the function of the conserved rare codons near the NS5B C-terminus, we mutated the rare leucine and valine codons at positions 9392, 9398, 9407, and 9413 (marked by triangle, square, diamond, and circle in Figure 5A) into common codons, i.e., codons that are translated with high efficiency. These changes always include the change of an inefficiently decoded Wobble base (here adenosine) to an efficiently decoded non-Wobble base (guanosine). Thereby, the rare Wobble leucine codon CUA is changed to the efficient leucine codon CUG, and the rare valine codon GUA is changed to the efficient valine codon GUG (Figure 5A, and compare Figure 3C). All these changes were applied while maintaining the resulting RNA secondary structure of the stem-loop 5BSL3.3 (Figure 5B, middle panel). Moreover, it is important to note that all these changes also occur in natural HCV isolates, but virtually only individually rather than in combination (Appendix A). As a control, we mutated the codons in a way that both the amino acids are changed to very similar but different amino acids, and the structure of the 5BSL3.3 is disturbed (Figure 5B, right panel). 

After transfection of these replicon RNAs into Huh-7.5 cells, HiBiT expression from all constructs (codon mutations and polymerase mutations, either individually or in combination) is very similar at 4 h and still at 24 h after transfection (Figure 5D). This indicates that translation efficiency from the transfected RNAs is very similar and not affected by the codon mutations. In conclusion, a change in codon efficiency at these positions has no significant effect on the overall efficiency of translation of the HCV proteins. 

However, at 72 and even more at 96 h after transfection, the overall HiBiT expression from the replicons with the original rare codons is significantly higher than HiBiT expression from the mutant carrying efficient (common) codons (Figure 5D). Thereby, replication active replicons are amplified nearly exponentially in the case of the rare codon context (more than 2-fold every 24 h), with weaker growth after mutation to efficient codons (Figure 5D). In contrast, HiBiT expression from the polymerase-deficient replicons declines to the background. These differences in HiBiT expression cannot be explained by subtle translation differences, which amplify over repeated replication cycles, since the translation efficiency from the mutated replicon (yellow in Figure 5D) is definitely not lower than translation from the wild-type replicon (at 4 and 24 h) but rather very slightly (albeit non-significantly) higher. The differences between the constructs already appear at 48 h after transfection with the wt polymerase replicons (Figure 5D). These differences in HiBiT expression cannot be explained by differences in cell number since WST-1 tests ("water soluble tetrazolium", which measure mitochondrial respiratory chain activity) do not show significant differences, and HiBiT expression values are corrected for slight variations in WST-1 readouts. 

Taken together, the mutations of the low-efficiency rare codons near the NS5B stop codon to highly efficient common codons make HCV RNA replication (but no translation) less efficient. 

## 3. Discussion

The HCV NS5B polymerase is translated as the last gene of the viral genome. Other proteins of the replication complex, e.g., NS4B, which is important for the recruitment of membranes as a housing for the replication complexes, and NS5A, which dimerizes [83,84] and acts as a coordinator for the replication complex and for assembly [8,25,26,27], are translated ahead of the replicase. The NS5B stop codon is followed only by the 3′UTR with a short variable region, a long flexible single-stranded poly(U/C) tract, and the conserved 3′X region that contains the minus-strand RNA synthesis initiation site at the genome’s very 3′-end. 

Here, we showed that rare, inefficient codons (mainly coding for leucine) are conserved at the end of the NS5B coding sequence, which serve to pause two ribosomes at the translation termination site and just upstream of it. We identified this ribosome pausing by ribosome profiling, a method that can be used to identify open reading frames, translation starts and stops, and ribosome occupancy on single mRNAs by the evaluation of translation starts and stops [85,86]. Thereby, on average, translation termination at mRNA stop codons does not give rise to particularly remarkable ribosome peaks in metagene profiles of cellular mRNAs [57,87]. Our transcriptome reads (Figure 2C) also show that sequencing bias is not the cause for the high ribosome peaks observed near the HCV stop codon (Figure 2B and Figure 3A). Therefore, we conclude that the two ribosome peaks seen at the NS5B stop codon and directly upstream of it are due to ribosome pausing. 

Mutation of the rare codons to efficient codons has significantly impaired expression of the HiBiT tag included in the viral replication protein expression cassette (Figure 5D). If this negative effect of the rare codons would primarily have implications on the translation level, translation efficiency of the mutated constructs would be different also at early time points after transfection (i.e., at 4 h after transfection, Figure 5D). This effect should occur with both NS5B replicase competent and deficient constructs. In contrast, the mutated (efficient) codons impair HiBiT expression only in replicating constructs (Figure 5D, at 72 and 96 h). This indicates that the (indirect) effect of the rare codons and the resulting ribosome pausing has implications for HCV RNA synthesis. 

Our analysis of NS5B codon sequences (Figure 3D and Appendix A) shows that these rare codons are conserved in most HCV isolates, although the use of common codons would be possible without disturbing the 5BSL3.3 structure (Figure 5B, middle panel). This supports the idea that these conserved rare codons play a role in the regulation of HCV replication. Such pausing is consistent with the previous finding that consecutive rare leucine codons can cause severe ribosome pausing in the presence of the natural cellular tRNA pool, whereas pausing can be relieved by adding additional tRNA [75]. This pausing event in the translation of HCV NS5B may allow the enzymatically active replicase core to find its genuine RNA template in *cis*, while the protein is still held in place by being stuck with its C-terminus in the exit tunnel of the paused ribosome. 

This action is supported by the intrinsic structure of the NS5B polymerase. The NS5B protein contains a stretch of about 55 amino acids at the C-terminus that is not part of the folded core of the polymerase but extrudes from one side of the enzyme core [80]. The 20 very C-terminal amino acids constitute a hydrophobic membrane anchor [88] by which NS5B is attached to the membranes in the replication complexes. The amino acids upstream of this membrane anchor can fold as a beta-sheet and occupy the active site cleft of the enzyme to inhibit its enzymatic activity [80,89]. However, this inhibitory beta-sheet is displaced from the active site during initiation [90]. Besides, the C-terminal hydrophobic membrane anchor is inhibitory to enzymatic action when the protein is expressed in vitro [80,89,90]. These findings point to an intrinsic safety mechanism, which would inhibit NS5B activity if it would be present in the cytosol in solution without being inserted into a membrane by virtue of its very C-terminal membrane anchor. 

NS5B can initiate RNA synthesis by a primer-independent mechanism [90]. It can bind single-stranded substrate RNA and slide along its template to accommodate the very 3′-end of the template in the active center [91,92]. The initiation complex composed of a single-stranded (ss) RNA template and an incoming nucleoside triphosphate is fragile and easily dissociates [93]. Even though single-stranded poly(C) is a much better template for NS5B than the genuine HCV plus-strand [94], NS5B can start de novo synthesis on a template containing the single-stranded right half of the 3′-terminal SL 1 of the HCV 3′UTR [95,96]. The above results suggest that the enzyme needs some time to accommodate its genuine HCV 3′-end template for RNA minus-strand synthesis initiation. Consequently, these data suggest a need for NS5B to be guided in *cis* to its genuine HCV RNA plus-strand 3′-end template present in close vicinity, in order to avoid the abortion of viral replication, as well as aberrant RNA synthesis initiation on non-cognate cellular RNA templates. 

The requirement for NS5B to find its RNA template in *cis* is also met by the characteristics of the genuine template. The NS5B coding region includes the CRE, which binds the NS5B polymerase protein at least as strong as the 3′X region [46,47,48,49]. Besides the ability of the CRE to stimulate translation [51], the RNA region containing the CRE also stimulates RNA minus-strand synthesis [97]. Although NS5B can start RNA synthesis on any RNA template [94], RNA synthesis initiation is largely stimulated by the presence of both the HCV 3′UTR X-tail, including the initiation site at the very 3′-end, and the 5BSL3.1, 3.2, and 3.3 as additional RNA elements, while it additionally requires the action of the NF90 and NF45 RNA chaperones [97]. 

In contrast to the fragile initiation event, once loaded onto its template, elongation of RNA synthesis by NS5B polymerase is highly processive. The corresponding complexes of NS5B with dsRNA are stable for more than 300 s [92], which is enough time to processively replicate the entire HCV genome by one NS5B molecule in one round without falling off. During HCV replication, the NS5B polymerase appears to produce only one minus-strand from each (surviving) incoming plus-strand [29]. This suggests that only one NS5B polymerase molecule needs to be functionally loaded onto the 3′-end of the plus strand. 

Our results in this study show that the translation of NS5B is slowed down in the C-terminal portion of NS5B. The ribosome placed on the termination site obviously takes time to complete termination since we see a strong ribosome peak at the termination site. This ribosome has one NS5B protein molecule still stuck in its ribosomal exit tunnel. Since the ribosomal exit tunnel shields about 30–40 amino acids from proteolytic degradation [98], we must assume that the C-terminal 30–40 amino acids of the NS5B polymerase are in an unfolded state in the ribosomal exit tunnel. In contrast, the upstream core of the enzyme can already properly fold [80] and thus is available for RNA substrate binding. 

Besides, the ribosome in the upstream peak may carry an already properly folded NS5B polymerase core. The valine codon in the active site of the upstream ribosome is 33 nucleotides upstream of the stop codon, i.e., 11 amino acids. Together with the 30 or 40 amino acids protected in the exit tunnel, this means that about 41 to 51 AAs are not allowed to fold together with the upstream portion of NS5B. In turn, at least the upstream protein up to amino acid 540 actually may fold. In the NS5B structure, the amino acids around position 540 are located in a stretch of amino acids that attach to the surface of the NS5B core [80] but may not be required for folding of the protein´s core. This implies that the core of the NS5B enzyme, which is required for substrate binding, can be assumed to properly fold, even when the C-terminal part of NS5B is still stuck on the slow valine codon in the upstream ribosome peak 2. 

From the above data and our own findings, we can hypothesize what the function of the ribosomal pausing at rare codons upstream of the HCV stop codon may be (Figure 6). The 3′-portion of the NS5B coding region contains the CRE/5BSL3.2, the 5BSL3.3 with rare codons, and the stop codon (Figure 6A). Upon translation, a ribosome can pass over the CRE and then be perhaps slightly slowed down first by the need to unfold the relatively strong stem of the 5BSL3.3. Then, the consecutive rare leucine codons in the 5BSL3.3 sequence essentially slow down the ribosome, which then pauses on the valine codon 9407 (Figure 6B). During this prolonged time, the NS5B protein core, which had already emerged from the ribosomal exit tunnel and which could fold into the active enzyme core, has enough time to find its genuine template in *cis*, i.e., the 3′-end of the HCV RNA in SL 1. The vicinity of the 3′X region with the paused ribosome is supported by the “kissing loop” interaction of the CRE/5BSL3.2 with the SL 2 of the 3′X region [39,55] and by the long single-stranded pol(U/C) tract acting as a flexible linker, consistent with the finding that a minimal length of the poly(U/C) tract is essential for replication [37,38,39]. 

We can only speculate if the 3′-terminal SL 1 or the CRE is bound first by emerging NS5B protein (Figure 6B); however, the distance between the CRE and the pausing position of the ribosome on the rare codons is small, perhaps not allowing the NS5B protein to bind to the CRE, whereas the 3′X region handed over by the long single-stranded poly(U/C) tract linker is flexible enough to accommodate the spatial requirements to allow NS5B binding to SL 1. When the ribosome that arrived first moved on to the termination site, again, this ribosome remains bound at that site for a considerable time while completing termination. At that time, a second ribosome encounters the rare codon region and is paused there (Figure 6C). It is not known for sure if the NS5B protein itself dimerizes, but the preceding NS5A actually can dimerize [83,84] and, by NS5A-NS5B interaction, may provide close vicinity for the two NS5B molecules. 

Finally, the NS5B molecule loaded to the HCV genome 3′-end has enough time to complete initiation and proceeds with RNA synthesis elongation (Figure 6D). In contrast, in the presence of efficient codons, the time for NS5B protein binding to its genuine RNA template is too short, resulting in lower RNA replication efficiency. However, it is important to note that these considerations are a hypothetical model based on our results shown here and on previous results from other studies, while elucidation of the mechanism still needs further investigation. The rare codons at the end of the NS5B coding region are conserved in most HCV isolates (Figure 3D and Appendix A), although the 5BSL3.3 RNA secondary structure also allows the use of efficient codons (Figure 5B, middle panel). Therefore, we consider that the proposed mechanism is active in most HCV isolates. However, it would be interesting to extend the present results to other HCV genotypes in future studies. 

Taken together, we showed that ribosomal pausing at rare codons near the end of the HCV NS5B replicase coding region is involved in efficient RNA replication, directly linking viral genome translation to the initiation of RNA synthesis. 

## 4. Materials and Methods

### 4.1. Plasmids

The JFH1/J6 chimeric full-length HCV genome (Jc1) was obtained from Ralf Bartenschlager [56]. The 5′UTR-core HCV replicon (Figure 3B) was assembled from a chemically synthesized backbone plasmid (synthesis by EuroFins, Ebersberg, Germany) and a fragment from a JFH1/J6 chimeric full-length HCV genome (plasmid pFK-JFH1-J6 C-846_dg (Jc1)_12961; in brief: “Jc1”) [56]. The backbone plasmid contained in 5′ to 3′direction: a T7 promoter, the HCV 5′UTR and core sequences (HCV positions 1 to 913), a linker sequence, and an encephalomyocarditis virus (EMCV) IRES sequence [99] to drive the translation of the HCV proteins. In the backbone plasmid, the EMCV IRES was followed by only small flanking regions of the NS3-NS5B sequence corresponding to the Jc1 clone, followed by the HCV 3′UTR, a hepatitis delta virus (HDV) antigenome ribozyme to generate an exact HCV genome RNA 3′-end, a T7 terminator sequence, and an EcoRI cleavage site. The main body of the wild-type (wt) NS3-NS5B sequence excised from the Jc1 clone was inserted into the backbone using BbvCI 233 nts downstream of the NS2-NS3 junction and AscI 151 nts upstream of the NS5B-3′UTR junction. In the SL I-III HCV replicon, the HCV 5′UTR and core sequences were replaced by the HCV 5′UTR sequence ranging from pos. 1 to only pos. 330, downstream of the HCV IRES SL III. The replicase-deficient plasmid variants were generated by the introduction of a GND mutation within the NS5B RdRp gene (318D->N) [81]. 

The sequence coding for the HiBiT tag (amino acids VSGWRLFLLIS), together with sequences coding for upstream (GGGSGG) and downstream (GGSGGG) glycine-serine linkers, was cloned into the HCV 5′UTR-core replicon construct between the codons for arginine (R) 543 and leucine (L) 544 in the sequence TPLPEAR-LLDLSS (NS5B AAs 537–549). This insertion point is located about 20 AAs upstream of the hydrophobic membrane anchor of the NS5B polymerase [80]. The second insertion point for the HiBiT tag is upstream of the NS3 gene segment coding sequence. Here, the coding sequence for the HiBiT AAs (VSGWRLFLLIS) was fused via a GGSGGG linker to the NS3 sequence (starting with AAAPITAYA…).

### 4.2. In vitro-Transcription

Replicon RNAs to be transfected were generated by in vitro-transcription from plasmid DNA templates linearized with EcoRI [59]. In vitro-transcription was performed using T7 RNA polymerase according to the manufacturer´s protocol with the following modifications: 3.5 mM of each ATP, GTP, CTP, and UTP, additional 5 mM MgCl_2_ and 10 mM DTT, 25 ng/µL of linearized plasmid DNA, and 1 U/µL of T7 RNA polymerase. After 3 h at 37 °C, another 0.5 U/µL of T7 RNA polymerase was added, and the transcription reaction incubated for another 2 h. Transcription reactions were frozen at −20 °C for at least 30 min to inactivate the T7 RNA polymerase, and template DNA was digested by 2 U RNase-free DNase I per 1 µg of DNA for 15 min at 37 °C. RNA was extracted with phenol/chloroform, and ethanol precipitated. Finally, RNA transcripts were dissolved in equal amounts of RNase-free water. 

### 4.3. RNA Transfection of Replicon RNAs and HiBiT Expression Assay 

RNAs were transfected into Huh-7.5 cells using Lipofectamine 2000 (ThermoFisher, Waltham, MA USA). For HiBiT expression experiments, about 24 h before transfection, Huh-7.5 cells were seeded in 24 well plates to 0.5×10^5^ cells/well with 0.5 mL 10 % DMEM. Transfection with Lipofectamine 2000 was performed at about 70% cell confluency. First, 0.75 µL of Lipofectamine reagent per well (3 µL Lipofectamine per µg RNA) and 125 µL serum/antibiotic-free DMEM were premixed, vortexed for 5 s, and incubated for 5 min. The 0.25 µg (0.1 pmol) of RNA solubilized in water was then added, and the mixture vortexed for 3 s. Mixtures were incubated at room temperature for 15 min, gently inverted 10 times, and carefully applied to cells dropwise. Cells were further incubated with complete DMEM for the times indicated.

WST-1 tests were started 45 min prior to a given time point, treating only 4 to 5 wells at a time to minimize cell stress. The medium was removed from wells, washed with 200 µL colorless DMEM (i.e., w/o phenol red), and WST-1 reagent solution (protected from light) was diluted 1:50 in 200 µL of colorless DMEM and added to the cells. The colorless or slightly red WST-1 tetrazolium salt substrate is converted to a red/yellowish formazan product by mitochondrial respiratory chain activity and serves as an indirect measure for total live cell mass in a well. After 30 min incubation, 100 µL of the supernatant was transferred to a 96 well plate, and WST-1 was measured in a Tecan microplate reader as absorption difference at 450 nm, using 650 nm as a reference. HiBiT expression was corrected for WST-1 measurements. 

Expression of the HiBiT tag was used as a measure not only for translation but indirectly also for genome replication since the amount of HiBiT expression represents the abundance of translation template RNAs and thus can be regarded as an indirect measure for RNA abundance. Detection of the HiBiT tag is very sensitive and very specific with low background [78,79]. Therefore, we relied on the HiBiT assay as a measure for translation and indirectly also for replication. For measuring HiBiT, the lytic reagent master mix (Promega, Madison, WI, USA) was prepared by diluting lytic buffer 1:1 with PBS. Plates were removed from the incubator 15 min before time, the medium was removed from cells, and cells were washed with PBS. Directly before application to the cells, the lytic reagent master mix was substituted with HiBit substrate (1:50) and LgBiT protein (1:100) (Promega) and gently mixed, and 70 µL of the resulting complete lytic reagent was applied to each well. The lysis was incubated under light protection on a shaker for 10 to 20 min. Lysed extracts were collected after repeatedly pipetting up and down, and the luminescence was measured in the luminometer. 

### 4.4. Statistical Analysis

Data are represented as means +/− standard deviations (SD, error bars). “n” indicates the number of independent experiments. Statistical significance was calculated by a two-tailed Student´s t-test (*: *p* < 0.05; **: *p* < 0.01; ***: *p* < 0.001), with a *p*-value < 0.05 considered significant. 

### 4.5. Transfection of HCV Full-Length Genomes and Ribosome Profiling 

Culture of Huh-7.5 hepatocarcinoma cells [100], in vitro transcription of HCV-RNA with the genotype 2a JFH1/J6 chimeric full-length HCV genome (JC1) [56], and transfection and further steps for ribosome profiling and transcriptome analysis were performed, as described previously [57]. In brief, Huh-7.5 cells were seeded in 15 cm dishes. Three replicates were carried out in parallel. One day after seeding, HCV-RNA was transfected. Six days post-transfection, cells were lysed for ribosome profiling and transcriptome analyses. Library preparation, ribosome profiling, rRNA depletion, and further experimental and bioinformatic processing were performed, as described previously; the method is described here in brief; for a detailed description, please refer to our previous study [57]. In parallel transfections, full replication of HCV in virtually all cells was checked by immunofluorescence against NS3, and HCV replication was also checked by Western blot against NS3 and core protein and by RT-qPCR against HCV plus strands [57]. Six days after the transfection of full-length HCV genomes, translation in the transfected cells was stopped by the translation elongation inhibitor cycloheximide. Cells were lysed in 500 µL lysis buffer (20 mM Tris–HCl (pH 7.5), 250 mM NaCl, 1.5 mM MgCl_2_, 1 mM DTT, 0.5% Triton X-100, 100 g/mL, 20 U/mL TURBO DNase) per 15 cm dish. The cell lysate was briefly cleared by centrifugation and then divided into two parts—one part used for ribosome profiling, and the other part used for transcriptome analysis. For ribosome profiling [57,101,102], lysates were treated with RNase I to digest RNA not covered by ribosomes, and the RNase I digest was stopped by the addition of a Superase inhibitor. These lysates were loaded onto 10–60% sucrose gradients [85]. Gradients were fractionated, and monomeric 80S ribosome peaks were identified by the UV 260 nm reading profile [57]. RNA was isolated from 80S peaks by phenol/chloroform extraction and precipitated with isopropanol. For total RNA analysis, RNA was isolated from cells using Trizol LS reagent and precipitated with ethanol. RNA was then fragmented under alkaline conditions [102]. After that, ribosome-protected and -fragmented total RNA samples were processed in the same way [57]. Three replicates were processed in parallel. Sample integrity was checked by an Agilent BioAnalyzer (Agilent Technologies, Waldbronn, Germany), and samples were sequenced by Next-Generation Sequencing (NGS) [57]. Reads were mapped to the chimeric Jc1 HCV genome. 

### 4.6. RNA Structure Prediction and Visualization

RNAs’ structures were predicted with RNAfold on the Vienna RNAfold WebServer [103]. For the visualization of predicted RNA structures, RNA sequences and Vienna dot-bracket outputs were loaded into the VARNA applet (v3-92) (http://varna.lri.fr) [104]. Base pair probability values were loaded into VARNA from the Vienna dot plot EPS file output and color code set to Vienna style. 

RNA double-strand (ds) probability (1–single strand probability) was calculated using RNAfold for 6 nts in the middle of windows with sizes of 150, 200, 250, and 300 nts moved in steps of 5 nts, and means over 25 nts were plotted over the nt number in the middle of the window. 

## Figures and Tables

**Figure 1 ijms-21-06955-f001:**
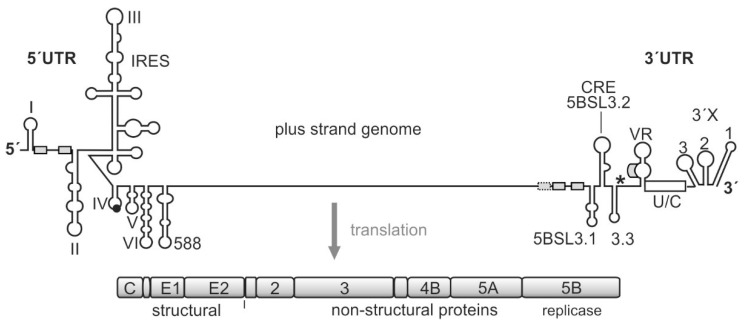
The hepatitis C virus (HCV) genome. The polyprotein open reading frame (ORF) is translated to a single polyprotein, which is co- and posttranslationally processed into the mature gene products, including the structural and the non-structural (NS) proteins. The gene segment for the NS5B replicase is located at the end of the polyprotein ORF. The HCV RNA genome contains several *cis*-elements, which are involved in the regulation of translation and genome replication [23,29,30,31]. In the 5′UTR, the internal ribosome entry site (IRES) element mediates translation of the polyprotein ORF. The stop codon is indicated by an asterisk. Several important RNA secondary structure elements, which are involved in HCV genome replication, are located at the very 5′- and 3′-ends of the genome. Other replication *cis*-elements are also located in the coding region, in particular, in the 3′-terminal portion of the NS5B replicase gene segment. These elements include the stem-loops 5BSL3.1, 3.2 (also called CRE for *cis*-replication element), and 3.3. microRNA-122 (miR-122) target sites are indicated by small boxes.

**Figure 2 ijms-21-06955-f002:**
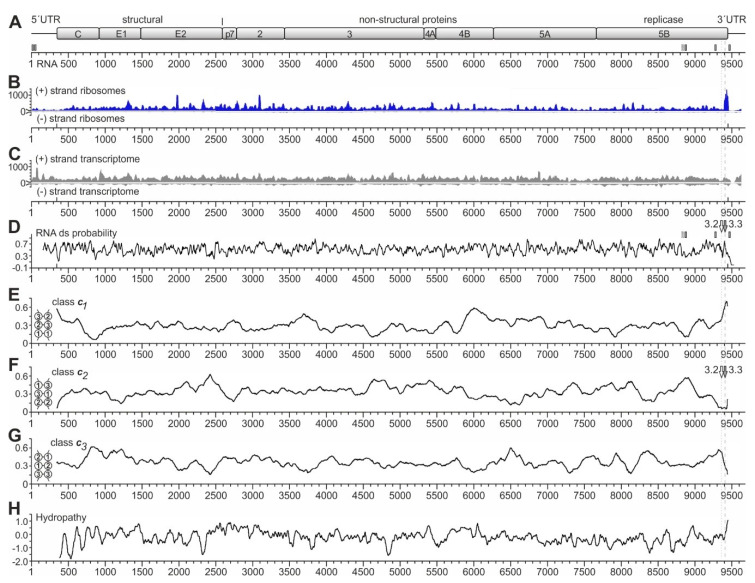
Ribosome profiling of the HCV RNA genome replicating in Huh-7.5 cells. (**A**) The HCV genome with the polyprotein coding regions for structural and non-structural (NS) proteins. The numbering shows nucleotide numbers of the Jc1 genome [56]. Two conserved miR-122 target sites in the HCV 5′UTR, sites 5B.2 and 5B.3 in the NS5B coding region, and the 3′UTR target site are shown as small boxes with solid lines, the non-conserved site 5B.1 as a box with dashed line [43,59]. (**B**) Ribosome profiling reads of the HCV plus (+) strand ribosomes and (−) strand ribosomes. (**C**) Transcriptome reads of the HCV plus (+) strand (upper panel) and (−) strand (lower panel). (**D**) RNA double-strand (ds) probability. miR-122 target sites in the 3′region are indicated. The positions of the stem-loops 5BSL3.2 and 5BSL3.3 are indicated by white and grey arrows, respectively, and dotted and dashed grey lines indicate their position relative to the ribosome peaks at the stop codon. (**E**–**G**) The frequencies of RNA secondary structure classes in which opposing codons would be required to pair (as shown in the inserts) were calculated for conserved HCV sequences [60]. For example, in class c1, position 1 of one codon would be required to pair with position 1 of an opposing codon, whereas in class c2, position 2 of one codon would be required to pair with position 2 of an opposing codon [60]. Windows of 200 nucleotides (nts) were moved in steps of 5 nts, and for each nucleotide, the means of class frequencies are calculated from all windows containing this given nt. As a result, the plots show if a certain RNA secondary structure class is under-represented due to selection pressure, which would suggest the presence of a functional RNA secondary structure [60]. Thereby, under-representation of RNA secondary structure class c1 indicates the general importance of codon position 1 to determine a specific amino acid, whereas under-representation of class c2 indicates the importance of codon position 2, which often determines hydrophobic amino acids in membrane standing protein regions [60]. In (**F**), the positions of the stem-loops 5BSL3.2 and 5BSL3.3 are indicated as in (**D**). (**H**) HCV polyprotein hydropathy plot [61], smoothened by plotting hydrophobicity means of sliding windows of 3 amino acids.

**Figure 3 ijms-21-06955-f003:**
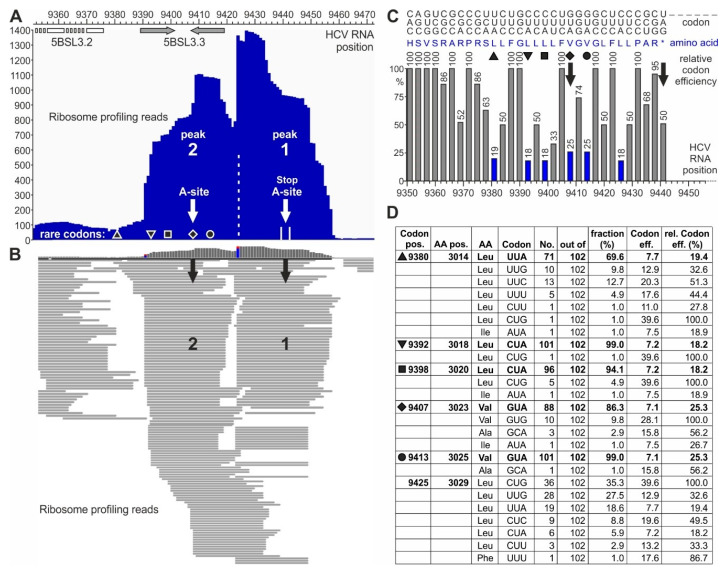
Pausing of ribosomes upstream of the HCV RNA stop codon. (**A**,**B**) Ribosome profiling reads on the HCV plus strand in the stop codon region. (**A**) Summarized reads of the peaks at the stop codon (peak 1) and one ribosome upstream of it (peak 2). White arrows indicate the center of each ribosome. The black diamond indicates the rare (low efficiency) valine codon in the center of the upstream ribosome peak 2, and other symbols (triangle, top-down-triangle, square, and circle) represent other rare efficiency codons (compare **C** and **D**). The positions of the stem-loops—5BSL3.2/CRE (white) and 5BSL3.3 (grey)—are indicated. (**B**) Single read resolution of peaks 1 and 2. Black arrows point to codons in the center of the ribosomes. For nucleotide resolution, please refer to Appendix A. (**C**) Codons with amino acids and human codon usage (graphical codon usage analyzer) [67]) of the HCV Jc1 genome in the stop codon region. Codons are shown on the top, depicted top-down (e.g., 5′-”CAC”-3′ for “H”, histidine, and 5′-”AGC”-3′ for “S”, serine). Amino acids are shown in blue in the middle. (**D**) Conservation of rare codons upstream of the HCV RNA stop codon among various HCV isolates. Codon usage was counted from 102 selected HCV isolates representing all HCV genotypes and subtypes [43]. The table shows how often (“No.”) a certain codon is used among the HCV isolates (“out of”). “Codon efficiency” is taken from the codon usage database for “Homo sapiens” [68] “rel. codon eff. (%)” is the respective codon efficiency (in %) calculated in relation to the most efficient codon for the corresponding amino acid. Black symbols indicate the rare efficiency codons as in (**A**,**C**). For detailed sequences, please refer to Appendix A.

**Figure 4 ijms-21-06955-f004:**
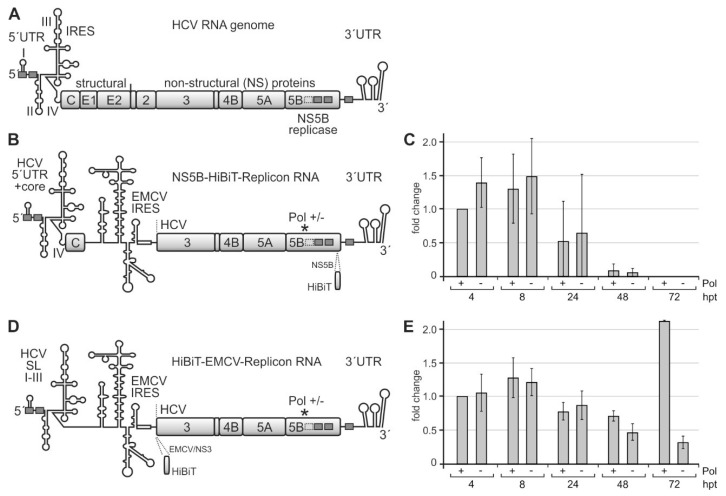
A system for measuring the efficiency of translation of the HCV non-structural proteins and of replication efficiency. (**A**) The HCV plus-strand RNA genome with the 5′ untranslated region (UTR) and internal ribosome entry site (IRES), the gene segments for structural and non-structural (NS) proteins, and the 3′UTR. Highly conserved miR-122 target sequences are shown as grey boxes with solid lines and a less conserved site with a dotted line. (**B**) The template replicon RNA for analyzing HCV translation and RNA synthesis contains the 3′-end of the HCV plus strand and the HCV replication cassette proteins NS3–NS5B. The encephalomyocarditis virus (EMCV) IRES drives the translation of HCV replication proteins. The active center of the HCV replicase NS5B is either wild-type (Pol +) or mutated to glycine-asparagine-aspartate, GND (Pol −). For allowing plus-strand production and by that amplification of the replicon, the 5′-end of the construct contains HCV 5′UTR and core protein-coding regions. Translation of the polyprotein was measured by expression of the HiBiT tag that is inserted near to the C-terminus of NS5B in this construct. (**C**) HiBiT expression from the construct shown in (**B**). Replicon RNAs were transfected into Huh-7.5 cells, harvested at times indicated (hpt, hours post-transfection), and the amount of HiBiT tag was measured. “Mock” background values from samples with no transfected RNA were subtracted from readings. (**D**) Modified replicon construct containing only stem-loops (SLs) I–III of the HCV 5′UTR but not the core coding region. In this construct, the HiBiT tag was fused to the coding region upstream of the NS3 gene segment, directly downstream of the EMCV IRES. (**E**) HiBiT expression from the construct shown in (**D**) at times indicated.

**Figure 5 ijms-21-06955-f005:**
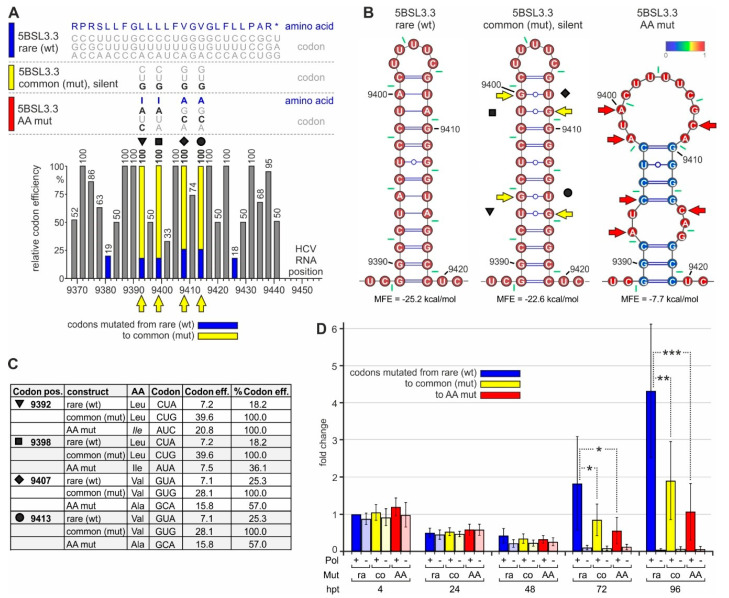
The impact of rare codons on NS5B translation and HCV replication. (**A**) The C-terminal region of the HCV NS5B replicase coding region with amino acids and codons on the top (like in Figure 2. The wild-type (wt) 5BSL3.3 sequence is shown on top (color code: blue), and below the silent mutations from rare to efficient (common) codons (color code: yellow) and amino acid mutations (color code: red). The panel below shows the rare codon and mutated codon efficiencies. Symbols (as in Figure 2, Appendix A) indicate the mutated codons (emphasized by yellow arrows). (**B**) Predicted RNA secondary structures of the 5BSL3.3 wt (left), the mutant in which rare efficiency codons were mutated to common codons (middle), and the amino acid mutations (right). Yellow or red arrows indicate mutated nucleotides. (**C**) Details of the codon mutations. (**D**) HiBiT expression from the replicon RNA shown in Figure 4D with the mutations shown in Figure 5A–C.

**Figure 6 ijms-21-06955-f006:**
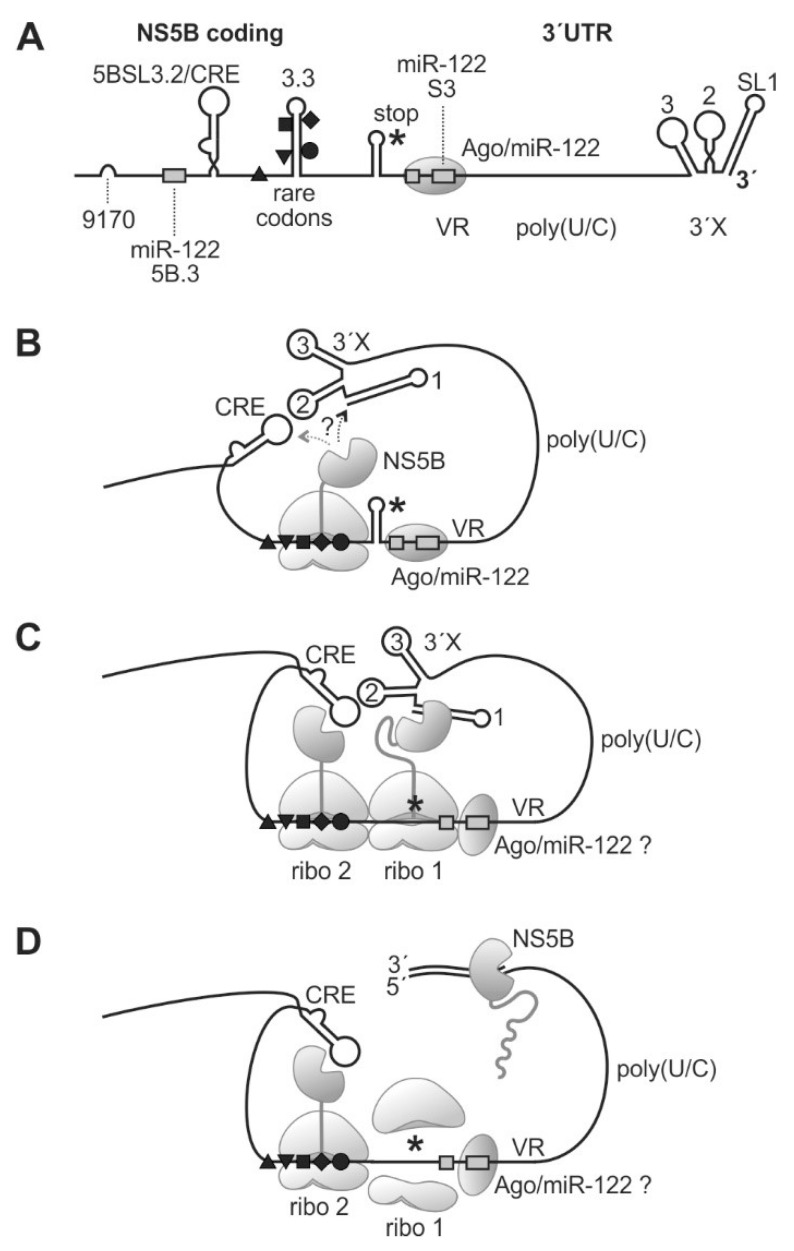
A model for ribosome pausing near the HCV genome stop codon and NS5B binding to the 3′-end of the viral RNA genome. The HCV 3′region (**A**) folds back to allow binding of yet incompletely translated NS5B protein to the SL1 of the HCV 3′end RNA (**B**). When completing NS5B translation (**C**), NS5B starts and proceeds RNA minus strand synthesis (**D**). Black triangles, square, diamond, and circle indicate rare codons as in previous figures. The NS5B stop codon is indicated by an asterisk. VR, variable region; 3′X, 3′X region with highly conserved stem-loops (SL) 1, 2, and 3.

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
