# Peer review of "Ribosome Pausing at Inefficient Codons at the End of the Replicase Coding Region Is Important for Hepatitis C Virus Genome Replication"

_ijms, 2020, doi:10.3390/ijms21186955_

Round 1
Reviewer 1 Report
Gesche et al. reported their study aimed to analyze the possible role of translation events at the end of the NS5B protein coding region and their possible role in regulating the initiation of RNA genome replication. The authors found ribosomal pausing at rare codons near the end of the HCV NS5B replicase coding region is involved in efficient RNA replication. The study is interesting and some minor concerns are raised.
- Authors may try to answer the question of whether the finding is related to the HCV specific genotype since only the JFH1/J6 chimeric full-length HCV genome (JC1) was used?
- I there any HCV quantification which did not describe in the methodology to demonstrate the HCV replication by the ribosome binding site?
- The limitations of the study have to be addressed. Ex. The data only provide the hypothesis of the ribosome regulation pathway with the mechanism still needing further studies.
Author Response
Reviewer # 1
Gesche et al. reported their study aimed to analyze the possible role of translation events at the end of the NS5B protein coding region and their possible role in regulating the initiation of RNA genome replication. The authors found ribosomal pausing at rare codons near the end of the HCV NS5B replicase coding region is involved in efficient RNA replication. The study is interesting and some minor concerns are raised.
Comment 1:
Authors may try to answer the question of whether the finding is related to the HCV specific genotype since only the JFH1/J6 chimeric full-length HCV genome (JC1) was used?
Reply:
We agree that this is an important question. Therefore, we have now included at the beginning of the discussion: "Our analysis of NS5B codon sequences (Figure 3D and Supplementary Figure S3) shows that these rare codons are conserved in most HCV isolates, although the use of common codons would be possible without disturbing 5BSL3.3 structure (Figure 5B, middle panel). This supports the idea that these conserved rare codons play a role in the regulation of HCV replication."
Moreover, we now state at the end of the discussion: "The rare codons at the end of the NS5B coding region are conserved in most HCV isolates (Figure 3D and Supplementary Figure S3), although the 5BSL3.3 RNA secondary structure also allows the use of efficient codons (Figure 5B, middle panel). Therefore, we consider that the proposed mechanism is active in most HCV isolates. However, it would be interesting to extend the present results to other HCV genotypes in future studies.".
Comment 2:
I there any HCV quantification which did not describe in the methodology to demonstrate the HCV replication by the ribosome binding site?
Reply:
This is an important question. We actually tried RT-qPCR to detect HCV replicon (better: "minus strand RNA synthesis template") RNA abundance in the transfected cells. However, we found that sensitive detection of low levels of these HCV replicon RNAs (in contrast to the high amounts of fully replicating HCV genomes in the ribosome profiling experiment) was hampered by high backgrounds resulting from contaminating transcription template DNA remnants, and the first results were somewhat less sensitive and more variable than the HiBiT readings, with the extent of variations making the results not really reliable without further trouble shooting. Therefore, in further experiments in the course of this study, we focused on the HiBiT assays as a routine assay for the low levels of replication of the HCV replicon constructs. Detection of the HiBiT tag is a measure for translation of protein from an intact RNA and therefore can be regarded as an indirect measure for RNA amounts, and it is very sensitive and very specific with very low background. Therefore, we relied on the HiBiT assay in the entire study. We have now included a corresponding explanation in the methods section: "Expression of the HiBiT tag was used as a measure not only for translation but indirectly also for genome replication, since the amount of HiBiT expression represents the abundance of translation template RNAs and thus can be regarded as an indirect measure for RNA abundance. Detection of the HiBiT tag is very sensitive and very specific with low background {Oh-Hashi, 2017;Schwinn, 2018}. Therefore, we relied on the HiBiT assay as a measure for translation and indirectly also for replication."
Comment 3:
The limitations of the study have to be addressed. Ex. The data only provide the hypothesis of the ribosome regulation pathway with the mechanism still needing further studies.
Reply:
We absolutely agree. Therefore, we have included in the discussion section - after describing the hypothetical model presented in Fig. 6 - the sentence: "However, it is important to note that these considerations are a hypothetical model based on our results shown here and on previous results from other studies, while elucidation of the mechanism still needs further investigation.".

Reviewer 2 Report
This study based on ribosomal profiling and the claimed development of a new approach for studying HCV translation suffers from several caveats, of which the main points are listed below:
Main issues
No information is provided regarding the way ribosomal profiling as been performed. Several types of information are lacking: features of the gradient used, gradient trace, HCV RNA signal across fractions, modes of purification of the translated ribonucleoprotein complexes.
The data showed herein suggest that none of the constructs are translationally / or replicatively active. Indeed, if one compares translational or replicative fold-changes of pastly developed replicons and other full-length HCV strains, one may observe that both viral parameters increase exponentially days after transfection.
The link between the study main message and the methods used and tools developed need be re-assessed.
Other point
Figure 1 is not necessary and more suited for a review paper.
Author Response
Reviewer # 2
This study based on ribosomal profiling and the claimed development of a new approach for studying HCV translation suffers from several caveats, of which the main points are listed below:
Main issues
Comment 1:
No information is provided regarding the way ribosomal profiling as been performed. Several types of information are lacking: features of the gradient used, gradient trace, HCV RNA signal across fractions, modes of purification of the translated ribonucleoprotein complexes.
Reply:
We apologize for stating these important methods too briefly, we agree that it may be important for readers to read an essential outline of the experiments in this manuscript. Therefore, we have now included a more detailed description of the ribosome profiling method in the Materials and Methods section 4.5.:
"… the method is described here in brief, for a detailed description please refer to our previous study (Gerresheim et al., 2019). In parallel transfections, full replication of HCV in virtually all cells was checked by immunofluorescence against NS3, and HCV replication was also checked by Western blot against NS3 and core protein and by RT-qPCR against HCV plus strands (Gerresheim et al., 2019). Six days after transfection of full length HCV genomes, translation in the transfected cells was stopped by the translation elongation inhibitor cycloheximide. Cell were lysed in 500 µl lysis buffer (20 mM Tris–HCl (pH 7.5), 250 mM NaCl, 1.5 mM MgCl2, 1 mM DTT, 0.5 % Triton X-100, 100 g/mL, 20 U/mL TURBO DNase) per 15 cm dish. The cell lysate was briefly cleared by centrifugation and then divided into two parts, one part used for ribosome profiling and the other part used for transcriptome analysis. For ribosome profiling (Gerresheim et al., 2019; Ingolia et al., 2012; Ingolia et al., 2009), lysates were treated with RNase I to digest RNA not covered by ribosomes, and the RNase I digest was stopped by addition of Superase inhibitor. These lysates were loaded onto 10 - 60 % sucrose gradients (Andreev et al., 2015). Gradients were fractionated, and monomeric 80S ribosome peaks were identified by the UV 260 nm reading profile (Gerresheim et al., 2019). RNA was isolated from 80S peaks by phenol/chloroform extraction and precipitated with isopropanol. For total RNA analysis, RNA was isolated from cells using Trizol LS reagent and precipitated with ethanol. RNA was then fragmented under alkaline conditions (Ingolia et al., 2009). After that, ribosome protected and fragmented total RNA samples were processed in the same way (Gerresheim et al., 2019). Three replicates were processed in parallel. Sample integrity was checked by an Agilent BioAnalyzer, and samples were sequenced by Next Generation Sequencing (NGS) (Gerresheim et al., 2019)."
Comment 2:
The data showed herein suggest that none of the constructs are translationally / or replicatively active. Indeed, if one compares translational or replicative fold-changes of pastly developed replicons and other full-length HCV strains, one may observe that both viral parameters increase exponentially days after transfection.
Reply:
Thank you very much for this comment! We agree that other well-known replicon RNAs (e.g. Lohmann et al., 1999, Science) can amplify strongly exponentially under antibiotic selection pressure. However, we must admit that we may have not made sufficiently clear in our manuscript that our "replicon" RNAs are not "replicons" in the generally used sense but they rather are "RNA synthesis template RNAs", since they do not contain a selectable marker.
Therefore, we have now supplemented the description of the replicons by the sentence: "It is important to note that the replicon RNAs do not contain a selectable marker like an antibiotic resistance gene. Thus, these RNAs rather constitute "RNA synthesis template RNAs" which are able to amplify by minus and plus strand synthesis but in the absence of a selectable marker. ".
In the legend to Figure 4C, we have supplemented the sentence: ""Mock" background values from samples with no transfected RNA were subtracted from readings." to make clear that we indeed see active translation of the RNA constructs.
However, the defective NS5B polymerase constructs show only translation, while the active NS5B polymerase constructs show a 2- to 3-fold increase every 24 hours (Figure 4E, from 48 to 72 hours, and Figure 5D, from 48 to 96 hours). Thus, we actually find exponential growth (seen in Figure 5D from 48 hours over 72 hours to 96 hours, nearly exponential growth with duplication every 24 hours), even though this may not be as strong as with classical antibiotic selected replicons.
Therefore, the interpretation of the results from Figure 4C was supplemented by the following phrase:
"HiBiT translation decreases at 24 hours after transfection and then drops to undetectable levels at 72 hours due to degradation of the transfected RNAs in the absence of active replication,".
The interpretation of the results from Figure 4E was supplemented by the following sentences:
"Within the first 72 hours, the expression of the HiBiT tag from the transfected RNAs did not yet show a strong exponential increase. However, those RNAs that carry an active NS5B polymerase catch up and start to replicate, which could be expected to continue at later time points (please see also below). In contrast, those RNAs that carry an inactivated NS5B polymerase (Lohmann et al., 1997) are degraded."
and the description of the results from Figure 5D was supplemented with: "Thereby, replication active replicons amplified nearly exponentially in the case of the rare codon context (more than 2-fold every 24 hours), with weaker growth after mutation to efficient codons (Figure 5D)."
Comment 3:
The link between the study main message and the methods used and tools developed need be re-assessed.
Reply:
We have now re-assessed our results in the light of the methods used. Therefore, we have now included an additional passage of text at the beginning of the discussion section:
"We identified this ribosome pausing by ribosome profiling, a method that can be used to identify open reading frames, translation starts and stops and ribosome occupancy on single mRNAs by evaluation of translation starts and stops (Stern-Ginossar et al., 2012; Andreev et al., 2017). Thereby, on average, translation termination at mRNA stop codons does not give rise to particularly remarkable ribosome peaks in metagene profiles of cellular mRNAs (Gerresheim et al., 2019; Andreev et al., 2015). Our transcriptome reads (Figure 2C) also show that sequencing bias is not the cause for the high ribosome peaks observed near the HCV stop codon (Figures 2B and 3A). Therefore, we conclude that the two ribosome peaks seen at the NS5B stop codon and directly upstream of it are due to ribosome pausing.
Mutation of the rare codons to efficient codons significantly impaired expression of the HiBiT tag included in the viral replication protein expression cassette (Figure 5D). If this negative effect of the rare codons would primarily have implications on the translation level, translation efficiency of the mutated constructs would be different also at early time points after transfection (i.e. at 4 hrs after transfection, Figure 5D). This effect should occur with both NS5B replicase competent and deficient constructs. In contrast, the mutated (efficient) codons impair HiBiT expression only in replicating constructs (Figure 5D, at 72 and 96 hrs). This indicates that the (indirect) effect of the rare codons and the resulting ribosome pausing has implications for HCV RNA synthesis."
Other point
Figure 1 is not necessary and more suited for a review paper.
Reply:
We agree that - on first glance - Figure 1 may appear a bit overdone in this context. However, during writing we realized that in the introduction and the results section (2.2), we often referred to RNA signals that could not be displayed in Fig. 2A, since there the drawing is to scale and leaves no space for detailing RNA signals in the UTRs or the CRE. Therefore, we would like to suggest to keep Fig. 1 as is, since it may be useful for readers looking at important RNA signals mentioned in the text (IRES, 5BSLs 3.1, 3.2, 3.3. and 3´X).
